# Remote Interactive Surgery Platform (RISP): Proof of Concept for an Augmented-Reality-Based Platform for Surgical Telementoring

**DOI:** 10.3390/jimaging9030056

**Published:** 2023-02-23

**Authors:** Yannik Kalbas, Hoijoon Jung, John Ricklin, Ge Jin, Mingjian Li, Thomas Rauer, Shervin Dehghani, Nassir Navab, Jinman Kim, Hans-Christoph Pape, Sandro-Michael Heining

**Affiliations:** 1Department of Trauma Surgery and Harald-Tscherne Laboratory, University Hospital Zurich, Ramistr. 100, 8091 Zurich, Switzerland; 2School of Computer Science, University of Sydney, Camperdown, NSW 2006, Australia; 3TUM School of Computation, Information and Technology, Technische Universität München, 85748 München, Germany

**Keywords:** RISP, remote interactive surgery platform, augmented reality, telementoring, AR-assisted surgery, MR-HMD, HoloLens2

## Abstract

The “Remote Interactive Surgery Platform” (RISP) is an augmented reality (AR)-based platform for surgical telementoring. It builds upon recent advances of mixed reality head-mounted displays (MR-HMD) and associated immersive visualization technologies to assist the surgeon during an operation. It enables an interactive, real-time collaboration with a remote consultant by sharing the operating surgeon’s field of view through the Microsoft (MS) HoloLens2 (HL2). Development of the RISP started during the Medical Augmented Reality Summer School 2021 and is currently still ongoing. It currently includes features such as three-dimensional annotations, bidirectional voice communication and interactive windows to display radiographs within the sterile field. This manuscript provides an overview of the RISP and preliminary results regarding its annotation accuracy and user experience measured with ten participants.

## 1. Introduction

Due to the continuous advances of telecommunication technologies, an increasing global interconnectivity has been achieved. Nevertheless, there remains huge discrepancies in large parts of the world regarding the distribution of medical education and surgical expertise [1]. “Remote surgery” systems, based on augmented reality (AR) and facilitated by increasing access to stable and high-speed internet, might be a remedy [2,3]. Such systems can utilize high-resolution cameras as well as augmented reality (AR) screens and allow for a variety of applications ranging from remote consultations to telementoring [4]. 

Telementoring describes remote guidance from an expert to a less experienced learner via telecommunication technology [5]. This is typically achieved using a live video feed and bidirectional communication. In the field of surgery, such technologies could be used to enhance the surgeons’ education efficiency by encouraging independence and to transfer expert knowledge in emergency scenarios or highly specialized cases which cannot be transferred [6]. 

In such cases, the use of AR technology offers additional benefits by the use of augmented annotations, which can be displayed on a mixed reality head-mounted display (MR-HMD) and which can then be viewed and manipulated by a surgeon within the sterile field [7]. In addition, critical information can be presented within the surgeon’s field of view (FoV) without the need to move the surgeon’s head [8].

Indeed, there have been several successful attempts to utilize AR to enable remote collaboration, for example, “Proximie”, “ARTEMIS” and “Rods&Cones” [8,9]. While these solutions already offer great benefits, they also have certain limitations with regard to visualization, accuracy or ease of use. “Proximie” and “Rods&Cones”, for example, are restricted by a loss of depth information as they show annotations indirectly through a 2D display [9]. MS Remote Assist [10], which enables annotation and augmentation of images in the real world through an MR-HMD, on the other hand, is limited by an insufficient accuracy of the annotations. ARTEMIS, which is a collaborative, mixed reality (MR), surgical telementoring system that allows the three-dimensional reconstruction of the operation site for a remote surgeon is limited by its complex and expensive setup, which requires seven external tracking devices and their calibration [8].

During the “Medical Augmented Reality Summer School” (MARSS) of 2021 and the subsequent competition, we set out to develop an AR-based platform called the “Remote Interactive Surgery Platform” (RISP), which should address these limitations by being:Easy to use and quick to set up;Able to accurately display three-dimensional augmented annotations without loss of depth information.

We decided to utilize the Microsoft (MS) HoloLens 2 (HL2), a state-of-the-art MR-HMD. The surgeon’s FoV is streamed to a remote consultant’s personal computer, and the remote consultant can support the surgeon through augmented annotations, medical images and voice communication. Possible applications of this system are:-To enable a quick evaluation from the senior consultant on call during complex emergency surgery;-As a means to provide medical expertise to remote areas which lack specialization for complex interventions when a transfer of the patients is impossible;-As a teaching tool for less experienced surgeons during routine operations.

This paper aims to introduce the technology, give an overview of the workflow during the MARSS competition and present our preliminary results regarding its annotation accuracy and user experience. The source code of RISP is available through a public code repository (https://github.com/Joon-Jung/Remote-Interactive-Surgery-Platform, accessed on 15 September 2022). 

## 2. Methods

### 2.1. Definitions

Operating surgeon: The person wearing the HL2 while performing surgery. The operating surgeon receives feedback/mentoring from the remote consultant via our software (see below) while communicating in real time.

Remote consultant: The remote consultant is using the software on his personal computer or tablet and sees the same as the operating surgeon sees through the HL2′s main camera. At the same time, the remote consultant can access radiographs and create annotations in the operating surgeon’s field of view.

### 2.2. Drafting and Development

During the initial meetings, we agreed to focus our project on the depiction of 3D annotations, which are to be displayed using the MS HL2. As we quickly realized that the built-in MS Remote Assist did not provide sufficient accuracy on curved surfaces, we decided to implement our own software. 

We tested the early implementation of our software with a setup in which we performed a simulated incision on an orange (Figure 1). Multiple attempts were made to address relevant issues, such as the annotations’ visibility on different backgrounds and retention of accurate depiction on the surface. We also added an augmented window to show the operating surgeon what the remote consultant is seeing as an additional awareness mechanism. 

### 2.3. Platform Requirements

During the following step of the planning stage, we defined the requirements for the platform in consensus meetings and by reviewing the literature [11]. These requirements were defined to ensure reliability during use in the above-mentioned applications. 

Three-dimensional projection of accurate annotations: The platform should enable the remote consultant to create annotations within a three-dimensional space; these annotations are then projected into the operating surgeon’s FoV. For our primary evaluation, we predefined a mean accuracy of at least 3 mm as the cut off. Larger inaccuracies would not allow proper identification of anatomic structures of the correct planes for surgical dissection. We chose this number based on the average sizes of veins, arteries and nerves, which can encountered during surgical dissection (e.g., the small saphenous vein is ~3.1 mm) [12].

Bidirectional voice communication: The platform is required to offer bidirectional voice communication. This ensures clear and direct interaction between the operating surgeon and the remote consultant. It gives the surgeon the opportunity to ask questions regarding any further steps [4].

Stability and reliability: A stable connection between the remote consultant and the operating surgeon is essential to ensure constant supervision and the ability to “intervene” at any given time [4]. Therefore, the platform should run reliably for at least 60 min without any major lag. In addition, the MR-HMD must be able to operate for said amount of time on a full battery charge. 

Responsiveness: Reliable and quick transmission of video, images and sound is needed for an intuitive communication. Thus, the video streaming should be lag free with minimal time delay. The remote consultant must be able to engage with the operating surgeon in real time to ensure the safety of their actions.

Ease of use and comfort: To facilitate intuitive interaction with our platform, all unnecessary visualizations should be hidden from the user interface. HL2, for instance, gives the users the “hand ray” to interact with the holograms. This is troublesome in our potential applications as it interferes with the visualization of more important information. Hence, the user interface has to be reduced to be as minimalistic and intuitive as possible. In addition, the MR-HMD should be comfortable to wear for long periods of time and should cause neither dizziness, vertigo, eye stress nor trouble concentrating [13].

Hands-free operations: Since the surgeon’s hands are sterile and occupied most of the time, it is essential that the system operates using voice commands. For more complex tasks within the interface (adjusting an augmentation’s position/size), it should allow a simple and sterile way of interacting without any additional devices [14].

Displaying complementary medical images within the surgeon’s FoV (X-rays, C-arm): The platform should be able to display various images in the surgeon’s FoV. These images should be moveable holograms and be placed in the surgeon’s FoV. This allows a broad and easily adjusted overview of additional information. Furthermore, orthopedic/trauma surgery often requires multiple intraoperative X-rays, which should be displayed as holograms in the surgeon’s FoV [14].

Measurement tool: The system should offer tools to measure different units. During surgery, oftentimes, various lengths, angles or diameters have to be assessed. For these tasks, an easy-to-access virtual measuring tool is advantageous. This can be accomplished by using different holographic devices, i.e., a ruler or goniometer [15].

Modifications: As a platform, RISP should be customizable for different uses and applications. The ability to implement new features in a user-friendly way helps to adapt the system to advanced and difficult cases.

### 2.4. “Remote Interactive Surgical Platform” RISP

#### 2.4.1. Functionality

The RISP was developed to be operative solely on two devices, HL2 and a personal computing device, as illustrated in Figure 2. These two devices are interconnected through a secured wireless connection. No additional equipment is required. From RISP on HL2 (RISP-HL2), the surgeon’s FoV video, voice and environment capture data are transmitted to the RISP on the personal computing device (RISP-Com.). Vice versa, the remote consultant’s annotation, voice and medical images are also transmitted. RISP supports telementoring through the following functions:Real-time streaming of the operating surgeon’s FoV and bidirectional voice communication;Annotations accurately augmented on the operating surgeon’s FoV in a three-dimensional (3D) space;Displaying additional medical images within the operating surgeon’s FoV sent by the remote consultant;Interacting with holograms and controlling RISP-HL2 through voice and hand gestures.

**Figure 2 jimaging-09-00056-f002:**
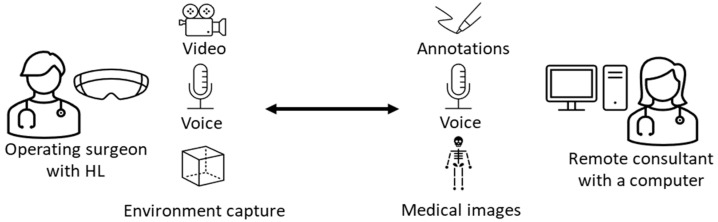
An overview of RISP with the data transmitted by the operating surgeon (video, voice and environment capture data) and the remote consultant (annotations, voice and medical images).

#### 2.4.2. Technical Implementation

RISP-HL2 was implemented using the Unity game engine (2019.04 version; https://unity.com/, accessed on 24 November 2021) and Mixed Reality Toolkit (version 2.7.2; https://github.com/microsoft/MixedRealityToolkit-Unity, accessed on 24 November 2021). Public libraries, including HoloLens2ForCV (https://github.com/microsoft/HoloLens2ForCV, accessed on 24 November 2021), HoloLensCameraStream (https://github.com/VulcanTechnologies/HoloLensCameraStream, accessed on 24 November 2021), HoloLens2-ResearchMode-Unity (https://github.com/petergu684/HoloLens2-ResearchMode-Unity, accessed on 24 November 2021) and MixedReality-WebRTC (https://github.com/microsoft/MixedReality-WebRTC, accessed on 24 November 2021), were utilized in the RISP-HL2 implementation. RISP-Com. consists of two applications, the voice communicator for enabling voice communication with RISP-HL2 and the main application for enabling all other functions. The voice communicator was implemented with the Unity game engine and the MixedReality-WebRTC library. The main application was implemented with the Python programming language and public libraries Open3D (http://www.open3d.org/, accessed on 24 November 2021) and OpenCV (https://opencv.org/, accessed on 24 November 2021). We implemented the transmission channel for the 3D annotation with the referencing source code of HoloLens2-Unity-ResearchModeStreamer (https://github.com/cgsaxner/HoloLens2-Unity-ResearchModeStreamer, accessed on 24 November 2021).

Real-time video streaming and bidirectional voice communication: RISP-HL2 captures the surgeon’s FoV using HL2′s main camera and sends it to RISP-Com. The captured FoV is transmitted as a series of images. The received images are sequentially displayed to the remote consultant as a video. The bidirectional voice communication is integrated through a bidirectional voice channel between RISP-HL2 and RISP-Com. with the WebRTC protocol (https://www.w3.org/TR/webrtc/, accessed on 24 November 2021).

Accurate 3D annotation in the operation surgeon’s FoV: RISP enables accurate 3D annotation through environment capture and finding the corresponding annotation locations with an environment reconstruction. RISP-HL2 records the operating surgeon’s environment with the HL2′s main camera and depth camera, used for sensing the 3D geometry of the environment. Then, RISP-HL2 produces two separate images, color and depth. The depth image is unprojected to a point cloud based on the pinhole camera model. The color image and point cloud are then transmitted to RISP-Com. The color image is shown to the remote consultant with an annotation tool. While the remote consultant is making the annotation on the color image, RISP-Com. reconstructs the environment by applying the ball-pivoting algorithm (BPA) to the point cloud [16]. After the annotation is finished, the corresponding location of the annotation is calculated by projecting the annotation onto the reconstructed environment. Then, the corresponding location is transmitted back to RISP-HL2 for the augmentation of the annotation. Finally, the 3D annotation is displayed to the operating surgeon.

Displaying complementary medical images: Medical images on the remote consultant’s computer can be displayed in the operating surgeon’s FoV. The RISP-Com. user interface (UI) allows the remote consultant to choose a medical image to send. After the selection, RISP-Com. transmits the medical image to the RISP-HL2. The RISP-HL2 displays the received image in a floating window in front of the operating surgeon’s FoV. Using gestures, the window can then be manipulated as desired (moving, scaling and removing). The sterile environment is not affected. 

Voice- and hand-gesture-based interaction: RISP-HL2 leverages the HL’s capabilities of voice recognition and full hand tracking. The control of RISP-HL2 is achieved by pressing holographic buttons and grabbing floating windows. For capturing the operating surgeon’s environment, RISP-HL2 records the operating surgeon’s voice. By saying “capture”, it triggers the environment capture. 

### 2.5. Validation and Testing 

#### 2.5.1. Conformation to Platform Requirements

The platform was tested in a multitude of simulated clinical scenarios throughout the development process. This included, among others, the calibration of touchless control in the sterile field by hand tracking and voice recognition, the placement of additional windows to allow visualization of radiographs, the implementation of bidirectional voice communication, the optimization of the client to remove distracting UI (Figure 1B) and, foremost, the repair of numerous stability issues. Once the software reached satisfying usability (Figure 3), we started the validation process in various experimental setups.

#### 2.5.2. Comparison of Annotation Accuracy with MS Remote Assist

Our first evaluation was to compare the RISP’s annotation accuracy with that of MS Remote Assist to validate the efficacy of our decision to use a custom software for the annotations. We did this by tracing straight 8 cm lines within the system and measuring the resulting annotation. This test was performed on a piece of paper, first, on a flat surface with four lines (1× horizontal, 1× vertical, 2× diagonal) and, subsequently, attached to a mannequin with two lines (1× horizontal, 1× vertical) to simulate curvature. The average distance between the straight and the traced line was calculated. 

#### 2.5.3. Evaluation of Setup, Stability, Voice Communication and Lag

The setup and stability were evaluated by measuring the time it took to establish a connection between the operating surgeon and the remote consultant using the RISP via the internet. Measurements were started once the HL2 was turned on and had reached its “home screen”. Average time for setup was calculated. The system was then kept running for 60 min before it was deactivated again. Software malfunction was defined as any loss of connection between the remote consultant and the operating surgeon. Voice communication was assessed by continuous verbal communication via the built-in headset and microphone. Lag was assessed with a stopwatch by placing the remote consultant and the operating surgeon in the same room to measure the time difference in information transmission between real time and the software.

#### 2.5.4. Practical Evaluation with Clinicians

As a next step, we evaluated the system with colleagues from the clinical setting. The purpose of this evaluation was to assess the accuracy of the annotations and to gain information on the user experience from novice end users who were not yet familiar with the application of the system.

Participants: Participants were chosen among the surgeons in training from the USZ’s department of traumatology. Ten clinicians were recruited for the evaluation. Every resident was eligible for voluntary participation.

Setup: The setup consisted of a 15 × 25 cm checkerboard pattern which was printed out, glued on a piece of foam rubber and then fixed to the flat surface of a table. The HL2 was fitted to the participants’ head, and they were briefly instructed on how to activate and use the platform. Then, participants were asked to trace a predefined pattern of annotations, which was projected onto the checkerboard, with a scalpel. The pattern consisted of 10 straight lines with different angles with varying lengths from 5 cm to 11.5 cm. Participants were asked to sit down but were allowed to position themselves as they were most comfortable. Cuts were to be performed with the dominant hand. The cuts in the paper were traced with a marker, and the average distance between the predetermined pattern and the traced line was calculated. 

User experience: After the procedure, participants were asked to complete a questionnaire to evaluate user experience. The questionnaire was self-administered and contained 18 statements that were to be graded from 1 (not at all) to 10 (absolutely). The statements inquired about the subjective user experience with regards to accuracy and consistency, the interactions with the augmented environment, the experience within the augmented environment and the potential uses of the RISP in a surgical setting. As no single standardized questionnaire was found in the literature that met our requirements, we decided to create a customized questionnaire. Some statements were modified from validated questionnaires such as the “System Usability Scale”. The complete questionnaire can be seen in the Results section (Section 3.3.2). 

## 3. Results

Disclaimer: The system presented in this section is still actively being worked on, and the software is currently in its prototype phase. Therefore, especially the measurements of accuracy and the evaluation of the user experience should be regarded as preliminary and will likely be subject to further improvements in the near future.

### 3.1. Comparison with MS Remote Assist 

The annotation accuracy of the RISP and MS Remote Assist is presented in Figure 4 using means and standard deviations (SDs). The RISP showed better average annotation accuracy on the flat (2.09 mm (SD: 0.76 mm) vs. 3.16 mm (SD: 1.59 mm)) and on the curved surface (2.21 mm (SD: 0.91 mm) vs. 7.32 mm (SD: 1.48 mm)). 

### 3.2. Setup, Stability, Lag and Voice Communication

Setup of the RISP system took, on average, 125 s (median: 107 s, SD: 76 s). While 4/5 setups were finished in under 2 min, we encountered a failed connection in one case, leading to a restart and a prolonged setup time of 258 s. Once the connection was properly established, it ran properly without loss of connection, drainage of the battery or major lag for the entire 60 min every time. Lag between real-world action and depicted action on the remote consultant’s screen was not perceivable and, therefore, could not be measured with a stopwatch. Voice communication worked unimpaired over the built-in microphone and speakers of the HL2. 

### 3.3. Evaluation with Clinicians 

A total of 10 participants were included for measurements of annotation accuracy and the user experience questionnaire. 

#### 3.3.1. Results of Annotation Accuracy 

Results from the evaluation of annotation accuracy are presented in Table 1. The accuracy was approximated by the average distance of the augmented and the traced lines. The overall mean distance from the original to the traced line was 1.55 mm, with a standard deviation (SD) of 1.61 mm. Maximal inaccuracies were approximated using the range from the minimal to maximal distance of the lines. This range was from 0 mm to 20.36 mm. Upon revisiting the individual results, we noticed that the annotated lines were occasionally cut off in the vertical direction and, therefore, shorter than intended. This led to large calculated inaccuracies in otherwise precisely matching lines. The remaining inaccuracies existed predominantly in the vertical plane (lines 1, 2, 7 and 10), as exemplified in Figure 5. 

#### 3.3.2. Results from User Experience Questionnaire

Table 2 presents the statements from the questionnaire and the median scores with range and SD. While the scores of the statements regarding the RISP’s technical implementation were in line with the findings from our accuracy testing, the statement results concerning the physical response to and individual handling of the AR showed a higher range and SD, which points to a larger variation between users. 

## 4. Discussion

With constant improvements of the technology, the use of AR-assisted surgery is slowly becoming more feasible, and new systems are constantly being developed [14]. During the MARSS 2021, we formed a cooperation of computer scientists (USYD) and clinicians (USZ), which allowed us to draft, develop and test an AR-based telementoring platform. While the development of our system is still ongoing, we established a stable version that allowed preliminary testing. Our results are as follows: The RISP worked as intended, and several of our predefined requirements could be implemented;The augmented annotations in the operating surgeon’s field of view showed an average accuracy of <2 mm. However, we still encountered occasional larger inaccuracies;Setup, stability and responsiveness of the platform were satisfactory;The user experience was overall positive; however, the personal response to experiencing the AR is subjective.

With the RISP, we aimed to create a toolbox which can assist a surgeon during an operation and which enables live and interactive collaboration with a remote expert. Our system was successfully used to capture and share the test environment with a remote consultant using the MR-HMD of the HL2. The remote consultant could then annotate information on the captured environment, which was augmented back to the corresponding location in the test environment. 

In this regard, our primary requirement of creating “Three-dimensional projection of annotations” was achieved. Our data further suggest that the average accuracy of the annotations was at a satisfying level (<2 mm). This improved accuracy of RISP in comparison to the similar MS Remote Assist was achieved by the utilization of the HoloLens’s raw depth image and ball-pivoting algorithm [16], which reconstruct surfaces of the physical world in high quality and enable the accurate placement of the user’s annotations to the corresponding locations in the real world. In contrast, MS Remote Assist depends on HoloLens’s built-in special mesh [17], which, as multiple studies [18,19,20,21] have noted, offers subpar surface reconstruction. 

In contrast to the average accuracy of the annotations, the consistency of annotation accuracy still requires improvement. This manifested in occasional larger inaccuracies. Especially, the accuracy in the vertical plane requires more adjustments. This is most likely due to the placement of the cameras on the surgeon’s forehead and, therefore, a certain vertical shift of the perspective in respect to the eyes. Furthermore, we observed occasional issues with lines being displayed incompletely, making them shorter than intended. These remaining inaccuracies could lead to potentially dangerous consequences in a clinical setting (i.e., when marking “danger zones” or vulnerable anatomic structures) and need to be addressed before even contemplating application in a clinical setting.

With regards to the other predefined requirements, our results suggest that “Stability and reliability”, “Responsiveness”, “Ease of use and comfort” and “Hands-free operations” were achieved. We were not able to manually measure the lag of the system due to a higher responsiveness than we could measure with a stopwatch in multiple runs. 

Our requirements of adding “Measurement tools” and other “Modifications” to the RIPS have not yet been achieved. 

Through the user experience experiment, we found that our system has reached a satisfactory level of accuracy and visibility, while the consistency of accuracy could still be improved. With regards to the individual handling of the augmented environment, the results were more diverse. We noted some variation in the way the participants interacted with and reacted to the augmented environment. This especially concerned discomfort, trouble concentrating and dizziness, which was expected as individuals have varying capacities to tolerate augmented and virtual environments [13]. These issues seem to grow stronger the longer one is exposed to the augmented reality. We have yet to perform experiments in which the operating surgeon is exposed to the AR for long periods of time, corresponding to a more complex surgery. Furthermore, performing additional experiments might identify factors which help or hinder an individual in tolerating the augmented environment. 

Overall, our results point to a system that shows promising potential but still requires extensive further adjustments, fine tuning and experimental testing. 

The implementation of AR- or MR-based systems to aid in surgery is a relatively new but rapidly expanding approach which was exemplified in a recent systematic review by Birlo et al. [11]. Especially in the field of orthopedic surgery, there have been numerous valuable contributions which evaluated the applicability of MR-HMDs in various settings [14]. These include the placement of pedicle screws in spinal surgery [22,23], component placement during arthroplasties [24,25] and the adjustment of intraoperative imaging modalities [26,27]. Overall, the review showed that most studies focused on the placement of tools or on image overlay for navigation, showing a clear focus on using AR-based systems to increase precision [11]. Indeed, most studies showed promising results with regards to the feasibility and the increase in precision; however, they were commonly concerned with one or a few isolated and very specialized tasks. 

Our system, on the other hand, was designed as a modular platform which is easy to use, quick to set up and which can be used in a multitude of settings, including in applications as a telementoring or teleconsultation system. The only requirements are a MS HoloLens, a computer and a stable internet connection. To our knowledge, this is the only AR-based application in this field which functions in this way. This is further substantiated by the recent systematic review of Jud et al., which showed that AR is, indeed, very rarely used as a telementoring tool in orthopedic surgery and that even fewer studies utilized the MS HoloLens [28]. While Condino et al. showed a promising approach for utilizing the MS HoloLens in a hybrid simulator for orthopedic surgery [29], we did not discover any literature describing an approach resembling the RISP. Nevertheless, one should consider that our system is not yet fully functioning as intended and that there are many future developments to be undertaken.

During the experiment with clinicians, we also conjectured that there would likely be certain reservations towards the use of new technology in the operating theatre, especially by more traditional, senior surgeons. Furthermore, the ethical aspects of testing such technology and the mandatory requirement of a certification for medical devices will require substantial efforts to overcome [30]. One possible way to increase familiarity with the use of AR in a surgical setting might be by using the “shadow surgeon” concept [31], a concept in which a colleague can follow an operation through the MR-HMD and later compare important aspects with the supervising consultant. Should their assessments of critical steps match, it would suggest that a sufficient presentation of the intraoperative setting can be achieved by using our system [31]. 

## 5. Limitations and Future Developments

Our study had several limitations. Most important was the preliminary nature of our results and the limited sample size. This was due to constant updates and improvements of our platform, requiring repeated validations for ensuring requirement satisfaction and the platform’s stability. We are also aware that we have not yet been able to show sufficient evidence that we have achieved all our predefined requirements. One major issue we are still facing is the inconsistency of the annotation accuracy, meaning the occurrence of occasional larger inaccuracies, which could be very dangerous in a clinical setting. These were, in part, due to a certain vertical shift between the operating surgeons’ eyes and the HL2′s sensors and, in part, due to annotated lines not being displayed at full length. We are currently in the process of fixing this issue. Once the platform’s consistency of annotation and its robustness have been improved, we are planning to perform extensive experiments with a larger cohort. The second limitation is that our experiments have so far been very limited and abstract. Ideally, our platform should be tested in mock scenarios in a cadaver lab with prepared injuries. As such setups are expensive, however, they will only be performed once sufficient annotation accuracy and feasibility have been verified.

## 6. Conclusions

During the MARSS 2021 and the subsequent months, we were able to draft, develop and evaluate an AR-based telementoring platform which combines 3D annotations and several additional features. Our initial aim of creating an AR platform which is easy to use and quick to set up, and which provides accurate 3D annotations, has been achieved.

The results presented in this article, however, are preliminary, and our system needs many further adjustments. This applies especially to the consistency of the annotation accuracy and the successful implementation of all our predefined requirements. Future improvements and experiments are planned.

## Figures and Tables

**Figure 1 jimaging-09-00056-f001:**
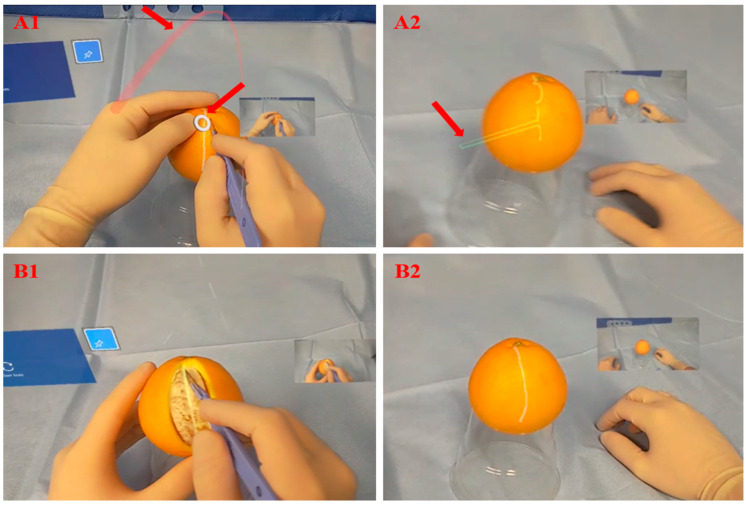
Images taken on the RISP during its development. (**A1**,**A2**) show an early version with distracting UI (red arrows) (**A1**) and poor depiction on the surface (**A2**). (**B1**,**B2**) show a later version where disturbing icons have been removed (**B1**) and the surface depiction is accurate (**B2**).

**Figure 3 jimaging-09-00056-f003:**
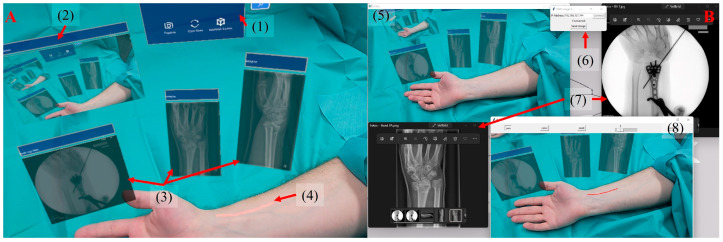
Visualization of a possible scenario for use of the RISP—(**A**) RISP-HL2 and (**B**) RISP-Com. The RISP-HL2 includes (1) control box, (2) preview of video, (3) medical images received from the RISP-Com., including C-arm imaging and (4) the projection of annotation. The RISP-Com. has (5) video streaming from the RISP-HL, (6) medical image sender, (7) sent medical images and (8) the color image with the annotation tool.

**Figure 4 jimaging-09-00056-f004:**
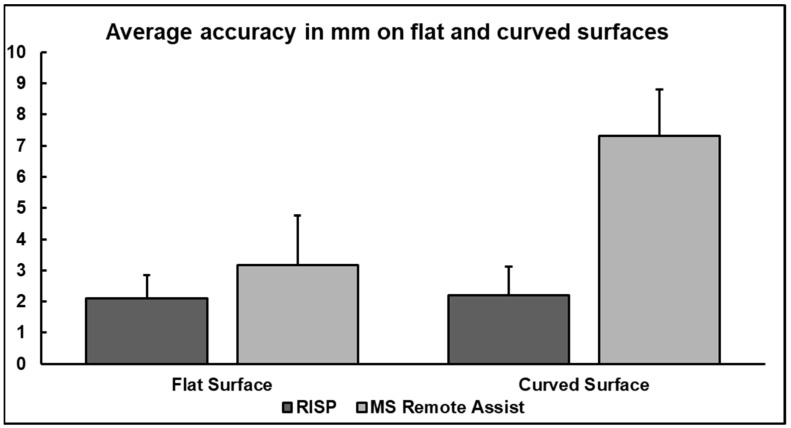
A bar chart visualizing the average accuracy of the RISP and MS Remote Assist on flat and curved surfaces. Values in mm. Error bars represent standard deviation (smaller = better).

**Figure 5 jimaging-09-00056-f005:**
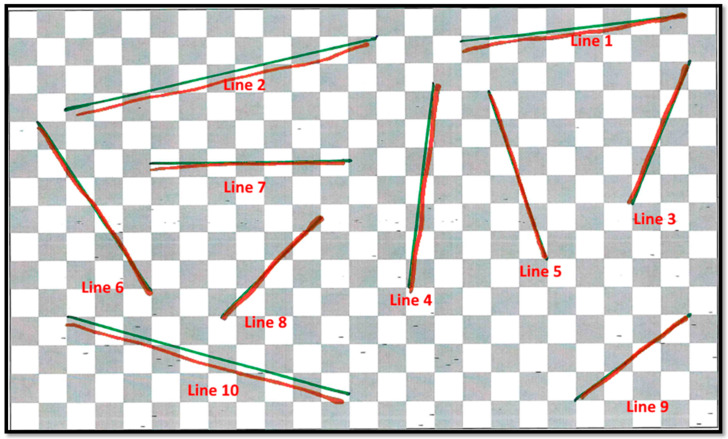
Result of annotation accuracy experiment with one participant. The traced lines (red lines) and the original lines (green lines) were combined through a registration after the experiment. Note that the inaccuracies existed predominantly in the vertical plane (lines 1, 2, 7 and 10).

**Table 1 jimaging-09-00056-t001:** Result of annotation accuracy testing presented in average distance, SD and range from minimal to maximal. The unit is millimeter.

Line	Inclination	Average Distance	SD	Range (Min.–Max. Distance)
1	7°	1.64	1.33	0–12.63
2	14°	1.49	1.28	0–7.66
3	68°	1.39	1.31	0–12.58
4	82°	1.52	1.40	0–11.90
5	70°	1.28	2.13	0–19.35
6	56°	1.15	1.07	0–9.34
7	0°	1.54	1.79	0–15.69
8	42°	1.35	1.06	0–11.85
9	36°	1.45	0.94	0–7.16
10	15°	2.25	2.37	0–20.36
Total		1.55	1.61	0–20.36

**Table 2 jimaging-09-00056-t002:** Mean score and SD to statements from the questionnaire. Score: 1 = not at all, 10 = absolutely.

Questions	Median	Range	SD
I felt the annotations were accurate.	8	7–10	0.99
I was consistently getting the same accuracy.	7	5–9	1.31
The annotations were clearly visible.	9	7–10	1.2
The augmented environment was responsive to actions that I initiated.	7	5–10	1.66
My interactions with the augmented environment seemed natural.	8	4–10	1.94
I felt proficient using the voice commands.	7	3–10	2.01
I felt that learning to operate the augmented environment would be easy for me.	8.5	6–10	1.51
I felt distracted by the augmented environment.	2	1–8	1.95
I had trouble concentrating.	2	1–8	2.1
I suffered from fatigue/headache/dizziness during my interaction with the augmented environment.	1	1–8	2.23
The experience hurt my eyes.	1	1–2	0.32
I suffered from nausea/vertigo during my interaction with the augmented environment.	1	1–2	0.32
I suffered from discomfort wearing the HoloLens.	3	1–8	2.15
Personally, I would say the augmented environment is practical.	8	3–10	2
I feel there are many different settings/operations in which the technology can be used.	9.5	5–10	1.6
I believe the RISP will be useful as a teaching tool.	8.5	4–10	2.15
I believe the RISP will be useful as a tool for remote consultations.	9.5	5–10	1.69
I believe the RISP will be a helpful tool during surgery.	8	5–10	1.25

## Data Availability

The described software is available here: https://github.com/Joon-Jung/Remote-Interactive-Surgery-Platform (accessed on 15 September 2022). Additional data is available on request from the corresponding author.

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
