# Peer review of "Remote Interactive Surgery Platform (RISP): Proof of Concept for an Augmented-Reality-Based Platform for Surgical Telementoring"

_2313-433X, 2023, doi:10.3390/jimaging9030056_

Round 1

Reviewer 1 Report (New Reviewer)

This is a preliminary work about the design and development of an AR-based remote interactive surgery platform. The system combines an AR headset with a computer to ease the communication between a surgeon in the operating room and a surgeon located elsewhere. The system allows the remote surgeon to follow the surgery and add annotations, voice comment, as well as show medical images. Simultaneously, the surgeon in the operating room can communicate with the remote colleague.

The work is certainly interesting, and I am impressed and appreciate that the idea has been designed at the MARSS and developed afterward in a multidisciplinary team of clinicians and computer scientists. However, the manuscript must be entirely rewritten, as in its current form is not suitable for a scientific journal. Below, some comments that I hope would help increase the quality and readability of the manuscript.

Title: It should clearly state that this is a preliminary study (e.g.: A proof of concept of an AR-based Remote Interactive Surgery Platform (RISP) for telementoring; or Remote Interactive Surgery Platform (RISP), a preliminary study of an augmented reality based platform for surgical telementoring)

Introduction: The introduction must be rewritten covering the following points:

-       Provide some context for the readers who are not familiar with some of the topics covered in the paper (i.e. AR, surgical telementoring)

-       Report background scientific data and discuss pertinent literature

-       Indicate a gap and state the need for your work

-       Show what you have done to address this need, clarifying what is new and why it is significant

At the beginning of the introduction, the authors state that “Despite continuous technological advances and increasing global interconnectivity, surgical expertise is still not equally distributed in large parts of the world” citing a paper which is ~15 years old. However, things have been changing constantly in the medical field; hence, I suggest finding a more recent reference.

Methods: Lines 74 to 88 (form “Drafting and development” to “exchange ideas and give feedback”) must be removed from the manuscript, as they are not relevant for the paper.

Lines 88 to 94 are part of the introduction rather than the methods

A critical part concerns the organization of the methods and results. In particular, one would expect to find the system description (i.e., software and communication system) in the methods section; instead, it is in the results. I foresee two possibilities: (i) re-structure the manuscript into: Introduction, System Design and Development (including system description), Tests (with discussion of the results); Conclusion. (ii) move the system development into the methods.

The authors state that they have added “an augmented window to show the operating surgeon what the remote consultant would be seeing” (line 103-104). Wouldn’t that be distracting?

Section 2.2.4 What does it mean that “Further simulations of clinical scenarios were performed to address the upper mentioned requirements and issues”?

Section 2.3.2. I have concerns about the methodology used to evaluate lag (i.e., using a stopwatch). Perhaps saving the timestamps of the data sent and received would help estimating the lag time? My concern is further supported by the comment of the authors in the discussion “Interestingly, we were not able to actively manually measure the lag of the system. This was due to a higher responsiveness than we could measure with a stopwatch in multiple runs” reported in the discussion.

Section 2.3.3. As the testing phase does not necessarily need clinicians to try the system, I strongly suggest the authors to increase the sample to have meaningful information which cannot be achieved with five participants. Also, user experience is usually assessed using validated questionnaires. Please, explain why you have decided to use a custom-made questionnaire rather than a standardized one.

Results: All the results in the text should be reported as mean and standard deviation/standard error (or median and range, when appropriate).

Section 3.5.1 what is min/max distance?

Section 3.5.2 Median and range are most appropriate for Likert scales, rather than mean and STDs. Also, the questionnaire should be reported into more details in the results section. For instance, why do you think the statement “the augmented environment was responsive to actions that I initiated” scored as low as 6.2 ± 2.07?

Discussion: what do the authors mean by the sentence: “Consistency of the annotations however, remains insufficient” (lines 339-340)

Consider discussing about the differences between the system you implemented and the existing ones.

Lines 372-376 “In regards to our requirement of enabling “Bi-directional voice communication, we could not devise a method to produce reproducible measurements to proof that it works as intended. We did however, not encounter any issue with the voice transmission during our multiple measurements. Considering the “Display of complementary medical images within the surgeon’s FoV”, we are in a similar situation” should be removed from the manuscript.

In particular, the way to compute the voice communication evaluation should have been defined prior to running the experiments.

As this is a preliminary work, I suggest adding a “Future Development” section.

Conclusion: the conclusion should: (i) Restate the problem statement addressed in the paper, (ii) Summarize your overall arguments and (iii) suggest the key takeaways from your paper. Please consider re-writing it accordingly.

Minors:

Page 2, line 49 VR is typically referred to a virtual reality system rather than an AR-virtual reality (VR) system, that is typically referred to as “AR”

I suggest putting the definitions (subparagraph 2.1) into a table or a textbox, removing them from the main text.

FoV (line 119) not defined in the text

I suggest to remove all the websites from the reference section and put them in the text (if  needed).

Section 2.3.1 I suggest refreshing the reader about the reason why you have compared your system with the MS remote assist

Table 3 is referred before table 1 and 2 in the text

Table 1 and Figure 4 show the same results, consider removing one of them.

Figure 5 shows how the experiment is performed rather than an actual result, so should be moved in the methods section.

Table 2 is a little bit hard to follow, a better representation would be using a plot showing the average data.

Author Response

Dear Reviewer,

We would like to thank you for this thorough and productive review. We addressed your comments and acquired measurements from five additional participants. We feel that the extensive revisions based on your comments substantially added to the quality of our manuscript and we hope that it now meets the standards for publication. In the following abstracts, we provide point-by-point replies to each of your comments:

Reviewer: “This is a preliminary work about the design and development of an AR-based remote interactive surgery platform. The system combines an AR headset with a computer to ease the communication between a surgeon in the operating room and a surgeon located elsewhere. The system allows the remote surgeon to follow the surgery and add annotations, voice comment, as well as show medical images. Simultaneously, the surgeon in the operating room can communicate with the remote colleague.

The work is certainly interesting, and I am impressed and appreciate that the idea has been designed at the MARSS and developed afterward in a multidisciplinary team of clinicians and computer scientists. However, the manuscript must be entirely rewritten, as in its current form is not suitable for a scientific journal. Below, some comments that I hope would help increase the quality and readability of the manuscript.”

Authors: Thank you. The manuscript has been extensively edited to address your comment wherever possible.

R: “Title: It should clearly state that this is a preliminary study (e.g.: A proof of concept of an AR-based Remote Interactive Surgery Platform (RISP) for telementoring; or Remote Interactive Surgery Platform (RISP), a preliminary study of an augmented reality based platform for surgical telementoring)”

A:  The title has been edited accordingly

R: “Introduction: The introduction must be rewritten covering the following points

-       Provide some context for the readers who are not familiar with some of the topics covered in the paper (i.e. AR, surgical telementoring)

-       Report background scientific data and discuss pertinent literature

-       Indicate a gap and state the need for your work

-       Show what you have done to address this need, clarifying what is new and why it is significant”

A: The introduction has been extensively edited to address these points. While the concept of telementoring is now explained in more detail, one can certainly expect that the readers of this journal are familiar with the concept of AR. We added more literature that is relevant and stated the research gap and our main intention more clearly.

R: “At the beginning of the introduction, the authors state that “Despite continuous technological advances and increasing global interconnectivity, surgical expertise is still not equally distributed in large parts of the world” citing a paper which is ~15 years old. However, things have been changing constantly in the medical field; hence, I suggest finding a more recent reference.”

A: We believe that this comment is now even more true than back then. We added a more recent reference to back this up.

R: “Methods: Lines 74 to 88 (form “Drafting and development” to “exchange ideas and give feedback”) must be removed from the manuscript, as they are not relevant for the paper.”

A: The indicated lines were removed.

R: “Lines 88 to 94 are part of the introduction rather than the methods.”

A: The lines were shifted to the appropriate section

R: “A critical part concerns the organization of the methods and results. In particular, one would expect to find the system description (i.e., software and communication system) in the methods section; instead, it is in the results. I foresee two possibilities: (i) re-structure the manuscript into: Introduction, System Design and Development (including system description), Tests (with discussion of the results); Conclusion. (ii) move the system development into the methods.”

A: We also agree to this comment and moved the system development into the methods.

R: “The authors state that they have added “an augmented window to show the operating surgeon what the remote consultant would be seeing” (line 103-104). Wouldn’t that be distracting?”

A: Regarding this comment, we must respectfully disagree: The window would certainly be distracting if it was fixed in the user’s focus, but as it can be placed in any position within the augmented environment, it can also be easily ignored. This point is reinforced by the results from our survey, in which the users don’t rate the augmented environment as distracting.

Also, the window is required to ensure what has been streamed to the remote surgeon by the local surgeon due to the user’s FoV is different from captured by the HoloLens’ camera.

R: “Section 2.2.4 What does it mean that “Further simulations of clinical scenarios were performed to address the upper mentioned requirements and issues”?”

A: Beside our initial testing on the orange, we also implemented numerous simulations and “mock scenarios” to simulate the use of the system in a clinical setting. We felt that elaborating on each of them would go beyond the scope of this article.

We changed the sentence to: “We consistently tested the platform in a multitude of simulated clinical scenarios throughout the development process”

R: “Section 2.3.2. I have concerns about the methodology used to evaluate lag (i.e., using a stopwatch). Perhaps saving the timestamps of the data sent and received would help estimating the lag time? My concern is further supported by the comment of the authors in the discussion “Interestingly, we were not able to actively manually measure the lag of the system. This was due to a higher responsiveness than we could measure with a stopwatch in multiple runs” reported in the discussion.”

A: Thank you for this great suggestion! We stated in the introduction that this manuscript aims to give an overview of our process during the development of our platform. Therefore, we report each of the methods as they were used. I wholeheartedly agree that the proposed method would have been better. However, not being able to record the lag with a stopwatch was a sufficient indicator for us that there were no major connectivity issues and we continued with the other aspects of our testing.

Once we have further advanced the development of our system, we will try get more specific on the actual lag time by using your proposed method.

R: “Section 2.3.3. As the testing phase does not necessarily need clinicians to try the system, I strongly suggest the authors to increase the sample to have meaningful information which cannot be achieved with five participants. Also, user experience is usually assessed using validated questionnaires. Please, explain why you have decided to use a custom-made questionnaire rather than a standardized one.”

A: We managed to increase the sample size to 10 participants and our results remained consistent. For future studies, we plan to include an even larger sample size, however we first aim to further improve our platform as described in the manuscript.

Regarding the user experience questionnaires, we chose to use a custom one, because no single standardized questionnaire we found in the literature met our requirements. We did however use modified questions from validated questionnaires such as the “system usability scale”.

R: “Results: All the results in the text should be reported as mean and standard deviation/standard error (or median and range, when appropriate).”

A: All the results are now reported in the appropriate manner

R: Section 3.5.1 what is min/max distance?

A: We calculated the closest distance from the original lines and to the traced lines by pixel-by-pixel after scanning the pieces of paper. The min/max distance refer to the highest inaccuracy of those pixels in comparison to the original lines. We have adjusted the result section to express this more clearly.

R: Section 3.5.2 Median and range are most appropriate for Likert scales, rather than mean and STDs. Also, the questionnaire should be reported into more details in the results section. For instance, why do you think the statement “the augmented environment was responsive to actions that I initiated” scored as low as 6.2 ± 2.07?

A: Results are now reported as median, range and SD. We chose to also include the SD because it gives additional information about the spread within the range. With the addition of 5 additional participants, the statement “the augmented environment was responsive to actions that I initiated” scored a lot higher and feel that the results from the questionnaire are now fairly unambiguous. However, we also provided more details in the results and the discussion.

R: Discussion: what do the authors mean by the sentence: “Consistency of the annotations however, remains insufficient” (lines 339-340)

A: Basically, we still encountered annotations that were too far off. We changed the text to make this clearer.

R: Consider discussing about the differences between the system you implemented and the existing ones.

A: We edited the discussion to elaborate on these differences:

R: Lines 372-376 “In regards to our requirement of enabling “Bi-directional voice communication, we could not devise a method to produce reproducible measurements to proof that it works as intended. We did however, not encounter any issue with the voice transmission during our multiple measurements. Considering the “Display of complementary medical images within the surgeon’s FoV”, we are in a similar situation” should be removed from the manuscript.

In particular, the way to compute the voice communication evaluation should have been defined prior to running the experiments.

A: We agree with this comment and removed this section. As we never encountered any issues with the voice communication during our experiments, we did not define a method to compute its evaluation.

R: As this is a preliminary work, I suggest adding a “Future Development” section.

A: Good idea. As many of our future developments are motivated by the limitations of the current study we added the section for future development to the limitations and went into more details of what is planned.

R: Conclusion: the conclusion should: (i) Restate the problem statement addressed in the paper, (ii) Summarize your overall arguments and (iii) suggest the key takeaways from your paper. Please consider re-writing it accordingly.

A: We have rewritten the conclusion accordingly.

Minors:

R: Page 2, line 49 VR is typically referred to a virtual reality system rather than an AR-virtual reality (VR) system, that is typically referred to as “AR”

A: We changed this to collaborative mixed-reality system as stated in the paper by Gasques et al.

R: “I suggest putting the definitions (subparagraph 2.1) into a table or a textbox, removing them from the main text.”

A: We put the definitions into a textbox.

R: “FoV (line 119) not defined in the text”

A: FoV is defined in the introduction.

R: “I suggest to remove all the websites from the reference section and put them in the text (if needed).”

 A: Agreed, websites were removed for the references if possible and relevant links to the utilized software was put in an additional textbox.

R: “Section 2.3.1 I suggest refreshing the reader about the reason why you have compared your system with the MS remote assist”

A: We added a sentence to refresh the reader about the reason we compared the systems.

R: “Table 3 is referred before table 1 and 2 in the text”

A: This is true; however, as this table is more relevant for the results we would prefer to keep it that way. We feel that adding an additional table for just the questions would be redundant. We hope you can agree with our opinion.

R: “Table 1 and Figure 4 show the same results, consider removing one of them.”

A: We removed Table 1

R: “Figure 5 shows how the experiment is performed rather than an actual result, so should be moved in the methods section.”

A: Figure 5 do not only show how the experiment is performed but it also shows how the results, especially regarding the accuracy in the vertical plane. We feel it fits better in the result section in combination with the new table 2.

R: Table 2 is a little bit hard to follow, a better representation would be using a plot showing the average data.

A: Agreed, we added a different more comprehensible table to illustrate the results. A plot of the average accuracy + SD would not represent the range of accuracies (this is especially important in regards to the measurements for max. inaccuracy.)

Again. Thank you very much for this very extensive and helpful review. We hope that we could provide sufficient revisions and that our revised manuscript now meets the standards for publication.

All changes in the manuscript are marked in red.

Reviewer 2 Report (New Reviewer)

This manuscript proposes an augmented reality based platform for surgical telemonitoring which is called Remote Interactive Surgery Platform (RISP). This developed platform enables an interactive real-time collaboration with a remote consultant by sharing the operating surgeon’s field of view through the Microsoft (MS) HoloLens2 (HL2).

In this paper, an over-view of the RISP and preliminary results regarding its annotation accuracy and user-experience measured with five participants are presented.

This manuscript is interesting, and its results can be considered as a contribution for improvement of remote interactive surgery platform. The presentation is good and contributes to the understanding of the method.

I think, the authors should be done the following corrections:

1.      Number of participants should be increased for meaningful results about usability of RISP.

2.      Comparison with similar studies in literature should be added to discussion in respect to usability, number of participants, etc.

Author Response

“This manuscript proposes an augmented reality based platform for surgical telemonitoring which is called Remote Interactive Surgery Platform (RISP). This developed platform enables an interactive real-time collaboration with a remote consultant by sharing the operating surgeon’s field of view through the Microsoft (MS) HoloLens2 (HL2).

In this paper, an over-view of the RISP and preliminary results regarding its annotation accuracy and user-experience measured with five participants are presented.

This manuscript is interesting, and its results can be considered as a contribution for improvement of remote interactive surgery platform. The presentation is good and contributes to the understanding of the method.

I think, the authors should be done the following corrections:

  1. Number of participants should be increased for meaningful results about usability of RISP.

  1. Comparison with similar studies in literature should be added to discussion in respect to usability, number of participants, etc.”

Dear reviewer,

thank you for your kind feedback. The manuscript was extensively edited. All changes in the manuscript are marked in red. We addressed both of your proposed corrections:

  1. We added the results from five more participants, making it 10 in total.
  2. We edited the discussion to reflect on similar studies in the literature

We hope that our revised version now meets the requirements for publication.

Thank you and all the best. 

Round 2

Reviewer 1 Report (New Reviewer)

The quality of the manuscript has overall improved. However, important changes must still be done, especially in the introduction, discussion and conclusion sections.

The introduction still needs improvements, as in its current form does not guide the readers through the understanding of the findings of the paper. The first sentence of the paper connects two points that are not related. I do see any relationship between the technological advances and increasing global interconnectivity, and the inequality of surgery in the world.

What is remote surgery? How is it currently exploited?

In the introduction you state that “there were several successful attempts to utilize AR to enable remote collaboration and describe them. However, in the discussion you also say that “Our system on the other hand is designed as a platform to be used in a multitude of settings, with a focus on telementoring, and to our knowledge, this is the only AR-based application in this field, which functions in this way”.

This is the structure I proposed:

1.     Provide some context for the readers who are not familiar with some of the topics covered in the paper (i.e. AR, surgical telementoring). This should be a general paragraph introducing your topic,  namely telementoring, current methods, and interest of AR-based system in the surgical theater.

2.       Report background scientific data and discuss pertinent literature. Describing in detail the existing solution and their limits

3.       Indicate a gap and state the need for your work. What is the need you want to address?

4.      Show what you have done to address this need, clarifying what is new and why it is significant”

Methods have definitely improved; however, I have a few comments left:

I believe textbox behave as any other figure and table in the text (i.e., with numbers and mentioned in the text)

In line 86 I suggest to add the following text after “our software”: “(see below)”, as it show the reader that further description of the system will follow.

Replace : with . at the end of the section.

Lines 98 – 102 can be rewritten as: “During the following step of the planning stage, we defined the requirements for the platform in consensus meetings and by reviewing the literature [11]”

Consider moving the textbox with the link in the supplementary materials for those interested.

The title “Experimental design” can be misleading as the experimental design is only the “practical evaluation with clinicians”. Consider calling it “Validation Tests”. Also, consider removing the section 2.3.3 and put that part in section 2.4. Section 2.4.3 can be called “Experimental Design”

To address my comment, you have said that you “chose to use a custom one, because no single standardized questionnaire we found in the literature met our requirements. We did however use modified questions from validated questionnaires such as the “system usability scale”. Please report this comment also in the manuscript.

Discussion: What do you mean by “The user experience is overall positive, however, the personal response to experiencing the AR is subjective.”? What can be a possible explaination?

The future development are mixed up in the discussion, and do not provide any insight on whether you have thought about possible improvements.

Conclusion: I don’t think it follows the proposed scheme yet.

(i)              Restate the problem statement addressed in the paper,

(ii)             Summarize your overall arguments

(iii)            Suggest the key takeaways from your paper.

What is the take home message of the paper?

Usually, as per requirement of the journals, the first table you mention must be called Table 1 and be the first one in the manuscript. However, if the journal let you do that. Also, table 1 does not exist, so change the number accordingly.

Author Response

Dear reviewer,

We agree that the manuscript has substantially improved during the revision process, and we would like to thank you again for the time and effort to review our manuscript. Your comments have been very helpful. For this second round we also tried to implement the comments wherever possible and reasonable.

Regarding the introduction however, we have to respectfully disagree with your position. We feel that a reader of this journal (and especially a reader of this special issue on medical augmented reality) should be familiar with the concepts of AR and remote surgery. We therefore do not see the need to extensively enlarge our introduction by describing concepts that we expect our readers to already know. The concept of telementoring on the other is clearly defined and explained in our introduction.

We also adapted our initial statement regarding the global interconnectivity to bring our point across more clearly.

Besides this, our introduction closely follows the proposed structure, as we introduced the scientific background, discussed the current gap in knowledge and propose a means to bridge this gap:

Considering the methods, all comments were addressed:

  • The textboxes were labeled, and the links were moved to the supplements and referenced in the text. 
  • Line 86 was edited as proposed.
  • Punctation was also edited as proposed.
  • Lines 98-102 were rewritten as proposed.
  • The title of section 2.4 was edited, and Section 2.3.3 was moved accordingly.
  • Also, we adapted the numbers for tables “2” and “3” to “1” and “2”.
  • We also added an explanation as to why we chose a custom questionnaire.

In the discussion, we modified the section regarding individual user experiences.

Regarding the future developments, it is correct that we mentioned them in the discussion several times, when we felt it was appropriate to reference planned future experiment. We do however also have separate section in which we discuss the limitations and how we aim to improve them in the future.

Regarding the earlier comment about other applications that utilize AR to enable remote collaboration, we modified the discussion to address this and point out more clearly the difference between our system and the existing ones.

Considering the conclusion, our take home message is simply, that we wanted to share what we achieved during the MARSS 2021 and that we are planning to further develop our software, until it might eventually reach a stage when it is usable in the clinical setting. We feel that this is sufficiently conveyed. We also restated the problems we addressed in our work, summarized our main findings and indicated that further experiments are planned. Therefore, we did not modify the conclusion any further.

All other changes to the manuscript are marked in red.

Again, thank you for all the helpful comments.

This manuscript is a resubmission of an earlier submission. The following is a list of the peer review reports and author responses from that submission.

Round 1

Reviewer 1 Report

This paper presents RISP, an AR platform for surgical telementoring. Authors position their platform as overcoming the issues of 2D annotations (non world-stabilized annotations on a video), or complex setups.

While I see the relevance of this study, the importance and the potential impact, the contributions are not solid nor clear, and there are some methodological flaws - that can be ok for a preliminary study, but need to be taken into account when making claims (or at least in limitations). Authors make claims that their data does not support, and, all in all, the novelty and benefits of RISP over existing systems is not demonstrated. For this reasons, which I detail below, I cannot recommend this submission for acceptance.

= Strengths

The use of AR and VR is growing in surgery for telementoring purposes, and we still do not have many studies that clearly identify the benefits of the approach, thus, this research is timely. There is a considerable amount of work in this paper, as authors implement and evaluate a system. One element that I really liked is the idea of showing to the local person the video that the remote person sees (figure 3-A-2). This serves as an awareness mechanism, which can improve communication (e.g., the local person does not refer to objects that are outside the field of view). This element could be one of the novel aspects of RISP.

= Weaknesses

1. Contribution

As I understand, this work presents two contributions: the RISP system, and its evaluation.

Regarding the RISP system itself, I cannot identify the novelty. The functions developed are already implemented in the system 365 Remote Assist: transmitting video & audio, pointing, and, annotating.

Thus, the issue with the first contribution is that it does not seem novel. If there are novel features, the authors need to make this more explicit.

Regarding the preliminary evaluation, authors compare RISP vs. 365 Remote Assist system to evaluate accuracy, also asking participants to assess easiness of use for RISP. The RISP system appears to be more accurate than the 365 Remote Assist. Although authors do not perform statistical tests, the tendency can be seen in the numbers authors present.

I see several issues with this second contribution:

i) If the goal was to increase accuracy, authors should discuss more how they implemented their algorithm that is designed to be more accurate. Is the expected increase in accuracy based on a more powerful scanning of the 3D world? Is it because they use an AI-based algorithm? In short, why should we observe an increase in accuracy?

ii) the results may lack validity, as they mix errors from the system tracking (what is being evaluated) and also by the person who performed the annotation (not relevant for the evaluation). It is hard to tear apart where does the increase in accuracy comes from, and this is something that authors need to consider in their data analysis.

iii) If the contribution is a more accurate system, it is not clear why they asked surgeon to perform this evaluation. The task was to trace lines, so anyone could have participated to the evaluation of accuracy. Regarding the usability questionnaires, only the last two items are related to medical work. All in all, I am not sure why surgeons were recruited.

Lastly, authors make claims for which they do not present support, for example "We were able to create an AR-based telementoring platform that is easy to use, quick to set up, stable and can be used in real time with little lag." There is no data collected for time to setup, stability or lag. Authors only evaluate accuracy and easiness of use.

2. Relevance and Validity of RISP requirements

RISP is based on a list of 9 requirements, which are unfunded (not based on literature, observations, nor a preliminary study). I understand that this list comes from the author's experience, since they are experts themselves. However, authors still need to justify this list. The most important is to make clear that these requirements respond to the task and constraints specific to surgery. There is a lot of literature authors can use to support this list.

Regarding the validity, although authors claim a "requirement validation and proofing" achieved by two institutions, there is little to no information on how this validation was conducted. To actually validate that the system fulfils these 9 requirements, authors would need to show proof that the system addresses them, one by one.

3. Organization

The paper can be better organized. Authors should follow the standard (and expected) structure of Methods, with headers that group information on Participants, Experimental Design, Apparatus, Task, Data Collection and Data Analysis

After much back and forth, I see authors conducted an experiment, where the independent variable is the type of software {RISP, MS 365 Remote aAssist} and the dependent variable is accuracy, measured as the mean distance between lines. The task involved tracing straight lines in different orientations. Still, organizing the paper in a standard manner would help the reader understand what the authors did.

= Minor

Figure 2 shows two different versions of the UI, where the first (left) has a UI that is more distracting than the second (right). However, the images are very similar, authors should better indicate what elements are distracting, why, or simply show two image where these elements are more clear.

Figure 4: what are the blue arrows?

Authors should consider using plots instead of tables, it is very hard to make inferences from numbers on a table. At least for the means, event if there is no statistical comparison, it helps to see visually the data to understand the results.

Reviewer 2 Report

This study presents the “Remote Interactive Surgery Platform” (RISP), an augmented reality (AR)-based platform for surgical telementoring. The study provides an overview of RISP and preliminary results regarding its annotation accuracy and ease of use measured with five participants. Overall, the presented RISP seems interesting and has potential. However, the study appear premature and the manuscript reflects this by an incomplete list of references, missing justifications, questionable research design, additional experiments needed, and a conclusion that are not supported by the results. Below please find specific comments to the manuscript.

The introduction does not include all relevant references. There are several more recent studies on the subject and a more recent systematic review available. To list a few:

1.       Birlo M, Edwards PJE, Clarkson M, Stoyanov D. Utility of optical see-through head mounted displays in augmented reality-assisted surgery: A systematic review. Med Image Anal. 2022 Apr;77:102361. doi: 10.1016/j.media.2022.102361. Epub 2022 Jan 12. PMID: 35168103.

2.       Bui DT, Barnett T, Hoang H, Chinthammit W. Usability of augmented reality technology in tele-mentorship for managing clinical scenarios-A study protocol. PLoS One. 2022 Mar 31;17(3):e0266255. doi: 10.1371/journal.pone.0266255. PMID: 35358249; PMCID: PMC8970358.

3.       Liu P, Li C, Xiao C, Zhang Z, Ma J, Gao J, Shao P, Valerio I, Pawlik TM, Ding C, Yilmaz A, Xu R. A Wearable Augmented Reality Navigation System for Surgical Telementoring Based on Microsoft HoloLens. Ann Biomed Eng. 2021 Jan;49(1):287-298. doi: 10.1007/s10439-020-02538-5. Epub 2020 Jun 5. PMID: 32504141.

4.       Rojas-Muñoz E, Lin C, Sanchez-Tamayo N, Cabrera ME, Andersen D, Popescu V, Barragan JA, Zarzaur B, Murphy P, Anderson K, Douglas T, Griffis C, McKee J, Kirkpatrick AW, Wachs JP. Evaluation of an augmented reality platform for austere surgical telementoring: a randomized controlled crossover study in cricothyroidotomies. NPJ Digit Med. 2020 May 21;3:75. doi: 10.1038/s41746-020-0284-9. PMID: 32509972; PMCID: PMC7242344.

5.       Wang S, Parsons M, Stone-McLean J, Rogers P, Boyd S, Hoover K, Meruvia-Pastor O, Gong M, Smith A. Augmented Reality as a Telemedicine Platform for Remote Procedural Training. Sensors (Basel). 2017 Oct 10;17(10):2294. doi: 10.3390/s17102294. PMID: 28994720; PMCID: PMC5676722.

Methods:

6.       Missing argument of why the annotations should have an accuracy of more than 0.3 cm to allow proper identification of anatomic structures and to define the correct plane for surgical dissection or the optimal point for screw placement.

7.       “We performed the first evaluation of the RISP’s annotation accuracy by comparing it to the MS remote assist by tracing four straight 8cm lines…”- Can the MS remote assist be perceived as the ground truth? - What is the accuracy of the MS remote assist?

8.       “User experience: After the procedure, participants were asked to complete a questionnaire to evaluate user experience.“ - Has the questionnaire validated?

Results:

9.       Why do you evaluate the average closest distance between the predefined straight line and the traced annotation? - What about the maximum distance, which must be very important for the system to be accurate?

10.   “Upon revisiting the individual results, we also noticed that lines were sometimes cut off in the vertical direction and therefore shorter than intended, leading to larger inaccuracies” – This issue should have been addressed before reporting the results.

Discussion:

11.   “The augmented annotations in the operating surgeon’s field of view showed a high average accuracy of less than 2mm. Nevertheless, the consistency is not yet sufficient for use in a clinical setup.” – The authors do not report if the augmented annotations were accurate according to the defined requirement of 0.3 cm.

Conclusion:

12.   “The RISP is an AR-based telementoring platform that enabled accurate 3D annotations.” The authors conclude that the annotation is accurate without proper evidence or statistical analysis.
